# Aberrant Resting-State Functional Connectivity in MDD and the Antidepressant Treatment Effect—A 6-Month Follow-Up Study

**DOI:** 10.3390/brainsci13050705

**Published:** 2023-04-23

**Authors:** Kangning Li, Xiaowen Lu, Chuman Xiao, Kangning Zheng, Jinrong Sun, Qiangli Dong, Mi Wang, Liang Zhang, Bangshan Liu, Jin Liu, Yan Zhang, Hua Guo, Futao Zhao, Yumeng Ju, Lingjiang Li

**Affiliations:** 1Department of Psychiatry, and National Clinical Research Center for Mental Disorders, The Second Xiangya Hospital of Central South University, Changsha 410011, China; 2Zhumadian Psychiatric Hospital, Zhumadian 463000, China

**Keywords:** major depressive disorder (MDD), resting-state functional connectivity, antidepressant, network-based statistics

## Abstract

Background: The mechanism by which antidepressants normalizing aberrant resting-state functional connectivity (rsFC) in patients with major depressive disorder (MDD) is still a matter of debate. The current study aimed to investigate aberrant rsFC and whether antidepressants would restore the aberrant rsFC in patients with MDD. Methods: A total of 196 patients with MDD and 143 healthy controls (HCs) received the resting-state functional magnetic resonance imaging and clinical assessments at baseline. Patients with MDD received antidepressant treatment after baseline assessment and were re-scanned at the 6-month follow-up. Network-based statistics were employed to identify aberrant rsFC and rsFC changes in patients with MDD and to compare the rsFC differences between remitters and non-remitters. Results: We identified a significantly decreased sub-network and a significantly increased sub-network in MDD at baseline. Approximately half of the aberrant rsFC remained significantly different from HCs after 6-month treatment. Significant overlaps were found between baseline reduced sub-network and follow-up increased sub-network, and between baseline increased sub-network and follow-up decreased sub-network. Besides, rsFC at baseline and rsFC changes between baseline and follow-up in remitters were not different from non-remitters. Conclusions: Most aberrant rsFC in patients with MDD showed state-independence. Although antidepressants may modulate aberrant rsFC, they may not specifically target these aberrations to achieve therapeutic effects, with only a few having been directly linked to treatment efficacy.

## 1. Introduction

Major depressive disorder (MDD) is an affective disorder characterized by depressed mood, somatic disorder, and cognitive dysfunction [1]. MDD has caused great concern as a public health risk and financial burden due to its high prevalence, high recurrence rate, and low remission rate [2,3,4,5]. Currently, antidepressants are the first-line treatment for MDD [6], while the overall cumulative remission rate is only 67% [7]. In addition, the choice of antidepressant for each patient is based on the trial-and-error strategy [8], which may lead to a series of adverse outcomes for patients such as delayed remission, greater symptom severity, poor compliance, etc. [9,10]. Therefore, it is essential to explore stable, objective, and measurable biomarkers for MDD [11].

In recent years, studies suggested that resting-state functional connectivity (rsFC) could help identify novel neuroimaging features in MDD with less heterogeneity [12,13]. Many previous studies have revealed aberrant rsFC in patients with MDD in the prefrontal cortex (PFC), orbitofrontal cortex (OFC), cingulate cortex, hippocampus, and amygdala at baseline [14,15]. In addition, recent studies have found that antidepressants can modulate rsFC in various brain networks [16,17,18]. However, it remains unclear whether aberrant rsFC is state-independent or state-dependent, as state-dependent rsFC is associated with the state of depression and may serve as an important mediator of MDD outcomes [19], while state-independent rsFC may reflect the susceptibility and underlie the biological basis of the onset and recurrence of MDD [20].

Since many studies observed that antidepressants seemed to normalize aberrant neuroimaging features observed in MDD [21,22], some researchers hypothesized that antidepressants may work by normalizing aberrant rsFC in patients with MDD [23,24]. However, results from previous studies have been inconsistent. Two meta-analyses found that lower baseline rsFC in patients with MDD (compared to healthy controls, HCs) overlapped with increased rsFC after antidepressant treatment [24,25]. Another study found that antidepressants did not normalize higher baseline rsFC in the medial PFC [26]. In addition, one study found that antidepressants can reduce large-scale network functional connections in MDD, but these connections were not significantly different from those of HCs [27]. Rütgen et al. found that rsFC between areas associated with affective (anterior insula) and cognitive (precuneus) empathy decreased in patients with MDD after treatment, with no significant difference noted in rsFC compared to HCs [28]. Moreover, Qin et al. found that only 30% of aberrant rsFC was normalized in remitters after antidepressant treatment [29]. Previous findings regarding the neurobiological mechanisms underlying the normalizing of aberrant rsFC by antidepressants in the treatment of MDD have been inconsistent and heterogeneous, and many studies have only investigated specific regions or networks instead of the whole brain. Therefore, there remains a need for further investigation into the effects of antidepressants on aberrant rsFC at the whole-brain level, in order to better understand the effects of antidepressants on aberrant rsFC and their potential role in the treatment of MDD.

To explore the possible biomarkers of treatment response and potential mediators of antidepressants, it is also important to identify any preexisting brain function characteristics related to the treatment response and differences in changes of brain function between remitters and non-remitters. Previous studies have examined the differences in baseline rsFC and the change from baseline rsFC to the post-treatment rsFC between MDD remitters and non-remitters during the acute treatment phase (6–8 weeks) [30,31,32,33,34]. However, it should be noted that many studies on antidepressant treatment have been short-term and have not thoroughly investigated the long-term outcomes. Hence, there is a need for further exploration of biomarkers that may indicate sustained non-remission in depression.

To address these issues, we used network-based statistics (NBS), a statistical tool that deals with multiple comparison problems at the level of interconnected sub-networks, which can increase specificity and sensitivity [35], to investigate the aberrant rsFC as potential biomarkers in MDD. Specifically, we aim to investigate: 1. the differences in whole-brain rsFC between patients with MDD and HCs at baseline and whether they display different rsFC changes with treatment; 2. the rsFC changes between baseline and follow-up, and the overlap between the aberrant rsFC at baseline and the rsFC changes at the 6-month follow-up; 3. the differences in rsFC at baseline between MDD non-remitters and MDD remitters, and the differences in pre- and post-treatment changes in rsFC between remitters and non-remitters.

## 2. Methods

### 2.1. Participants and Clinical Characteristics

Two hundred and thirty-five patients with MDD were recruited from the inpatient and outpatient clinical departments of the Zhumadian Second People’s Hospital from 2013 to 2018. The inclusion criteria for the patients were as follows: (1) age between 18 and 60 years old, (2) able to consent and complete the assessments in our study, (3) right-handedness, (4) diagnosed with MDD according to the Diagnostic and Statistical Manual of Mental Disorders, Fourth Edition (DSM-IV), (5) off psychotropic drugs for at least 2 weeks (6 weeks for fluoxetine), and 6) scored at least 20 on the 24-item Hamilton Rating Scale for Depression (HAMD-24). Exclusion criteria for the patients were as follows: (1) current or previous comorbid psychiatric disorder diagnoses in DSM-IV, (2) history of cerebral injury, coma, epilepsy, severe physical disease, or drug abuse, (3) current serious suicidal ideation or suicide attempts, (4) current pregnancy or lactation, (5) color blindness (unable to complete the neurocognitive tests), (6) having used anticoagulants (heparin or warfarin, etc.), glucocorticoids, or received treatment for thyroid diseases in the past 3 months, (7) having received any neurocognitive assessments similar to this study in the past 12 months, (8) positive urine drug screening results or abnormal thyroid function test, (9) any contraindication for MRI. One hundred and sixty-three HCs with matched sociodemographic characteristics were recruited from nearby communities and rural areas. The inclusion criteria for HCs include meeting criteria 1–3 of the MDD inclusion criteria, as well as the following: (1) scored less than or equal to 7 on HAMD-24; (2) gender, age, and years of education matched as closely as possible to the patient group. Exclusion criteria for the HCs were any history of psychiatric disorders or family history of psychiatric disorders according to the DSM-IV criteria, as well as any exclusion criteria applied to the patient group. During the 6-month antidepressant treatment, patients who were lost to follow-up, asked to withdraw, reported any active suicidal ideation or action, received ECT or rTMS, or presented symptoms which met other diagnostic criteria in the DSM-IV were further excluded. This study was approved by the Medical Ethics Committee of the Zhumadian Second People’s Hospital and the Second Xiangya Hospital of Central South University, and all participants signed informed consent.

### 2.2. Study Design

Demographic information, clinical characteristics, and depression and anxiety severity assessments (using HAMD-24 and Hamilton Anxiety Scale, HAMA) were collected at baseline. Participants then completed the structural MRI (sMRI) session and rs-fMRI scans. After baseline assessment, MDD participants received the first-line antidepressant treatment under clinician guidance during the study period. Augmentation medications, hypnotics, and sedatives were also used as appropriate based on clinical judgment. Depression and anxiety severity of patients with MDD were assessed using HAMD-24 and HAMA, respectively, at the end of the 1st, 2nd, 3rd, 4th, 5th, and 6th months. Of the initial 235 patients with MDD, 6 patients who had a changed diagnosis, 11 patients who received ECT or rTMS, 10 patients with incomplete MRI data, and 12 patients with excessive head movement (mean frame to frame displacement, FFD, greater than 0.2 mm) were excluded. A total of 196 patients with eligible baseline MR scans were included in the study. Of the initial 163 HCs, 6 were excluded due to excessive head motion and 14 were excluded due to incomplete basic information or incomplete MRI data. In total, 143 HCs were included in the analysis. Among the 122 patients who completed all clinical assessments and the second rs-fMRI scan after 6 months, 8 were excluded due to excessive head movement, resulting in 114 patients being included in the follow-up analysis. Clinical remission was defined as a HAMD-24 score ≤ 7 at the 6-month follow-up.

### 2.3. MRI Data Acquisition, Preprocessing and Network Construction

Details of MRI data acquisition were published previously [36,37] and can be found in Appendix A. In short, MRI data for both visits were collected on a 3T GE scanner (Signa HDxT 3.0T) using a 16-channel head coil. The T1 images were normalized to the Montreal Neurological Institute (MNI) space to obtain the transformation matrix, and the functional image was slice-time corrected and motion-corrected. Functional images with a mean frame to frame displacement (FFD) greater than 0.2 mm were excluded from further analyses. The functional image was then co-registered to the structural image. Nuisance signals, including linear, quadratic, cubic drift, a 24-parameter model of motion, the mean cerebrospinal fluid signal, and the mean white matter signal, were regressed out and the functional image was then temporally smoothed by a low-pass Gaussian filter (0.12 Hz).

Next, we warped the 268 regions of interests (ROIs) atlas [38] back to the individual subject spaces using the inverse transformation matrix in the T1 normalization step, and then applied this atlas to individual fMRI scans to extract 180 time-series from each ROI. Time-series of 12 brain regions were excluded due to missing fMRI signals. The Pearson correlation coefficient r of all the ROI time series pairs (256 × 256) was calculated and converted into Z scores using the Fisher Z-transformation, forming a 256 × 256 correlation matrix containing 32,640 unique rsFC for each participant.

### 2.4. Network-Based Statistics Analysis

The present study employed NBS to test four null hypotheses. Firstly, we hypothesized that there would be no significant differences in baseline rsFC between patients with MDD and HCs. Secondly, we hypothesized that there would be no significant differences between baseline and 6-month rsFC in patients with MDD. Thirdly, we hypothesized that there would be no significant differences in baseline rsFC between MDD non-remitters and remitters. Lastly, we hypothesized that there would be no significant differences in changes of rsFC from baseline to 6 months between MDD non-remitters and remitters. To test hypotheses 1 and 3, analysis of covariance (ANCOVA) was used in NBS, with age, gender, education, and head motion included as covariates. For hypothesis 2 and 4, repeated measures ANCOVA (RMANCOVA) was used in NBS, using the same covariates as ANCOVA. The *p*-values of these hypotheses were initially tested on each of the 32,640 unique rsFC. To ensure the significance of the results, the *p*-value was set to 0.001 to reject H_0_ to obtain a set of suprathreshold connections for subsequent comparisons. A breadth-first search was used to identify any connected sub-networks that may exist in the set of suprathreshold connections [39]. Next, 5000 permutation tests were performed to calculate *p*-values for each potentially significant sub-network that could reject H_0_. Specifically, in each permutation, the group to which each subject belonged was randomly swapped. The maximal component size was then stored in a set of suprathreshold connections derived from each of the 5000 permutations. Differences between groups were calculated using the formula *P* = K/5000, where K represents the total number of permutations in which the maximal component size is larger than the actual size, and a *p* < 0.025 indicates a significant difference in rsFC between groups.

### 2.5. Overlap of Functional Connectivity Sub-Network

Hypergeometric cumulative density function was used to determine the significance of overlapping networks, which calculates the probability of drawing up to x of K possible items in *n* drawings without replacement from an M-item population. We implemented this function in MATLAB as follows: *P* = 1 − hygecdf (x, M, K, N), where “x” equals the number of overlapping rsFC, “K” represents the level of rsFC that significantly decreased or increased compared to HCs, “N” represents the level of rsFC that significantly increased or decreased over time in patients with MDD, and “M” represents the total level of rsFC (32640).

### 2.6. Statistic Analysis

Except for the neuroimaging analysis, all the statistical analyses were performed using SPSS 24.0. The Kolmogorov–Smirnov test (K–S test) was applied to identify whether the demographic and clinical data were normally distributed. We performed independent sample t-tests for between-group comparisons of continuous variables with normal distribution, analysis of variance (ANOVA) for multiple group comparisons of continuous variables, Mann–Whitney rank sum test for continuous variables without normal distribution, and chi-square tests for categorical variables. We used RMANCOVA to compare the rsFC at baseline and at the end of the 6-month follow-up, with age, gender, years of education, and head motion (mean FFD) of MRI data at baseline and 6-month as covariates. False discovery rate (FDR) correction was applied for multiple comparisons. Results were considered significant at *p* < 0.05 (two-tailed).

### 2.7. Visualization of the Network Anatomy

We used Bioimage Suite (https://www.nitrc.org/projects/bioimagesuite/, accessed on 9 March 2022) to visualize our results. Brain circle maps and functional connectivity maps were generated to visualize significant rsFC differences in the NBS analysis results. In the brain circle map, the 256 brain regions were grouped into 10 macroscopic brain regions, including prefrontal, motor, parietal, temporal, occipital, limbic cortex, insula, cerebellum, subcortical, and brain stem. Each macroscopic region was assigned a different color, and connections between each brain region were depicted as lines. In the brain functional connectivity map, lines represent rsFC between brain regions, and spheres represent brain regions, with a larger sphere indicating a greater level of rsFC in the brain region.

## 3. Results

### 3.1. Demographic and Clinical Characteristics

A total of 196 patients with MDD and 143 HCs were included in baseline analyses. Among the 114 patients included in the follow-up study, 83 (72.8%) achieved clinical remission, while the remaining 31 (27.2%) did not. There were no significant differences in demographic and clinical characteristics between the baseline MDD group and the HC group, the follow-up MDD group and the HC group, or the follow-up MDD group and the follow-up excluded group (Table 1). Additionally, there were no significant differences in demographic and baseline clinical characteristics between the remitted group and the non-remitted group (Table 2).

### 3.2. Characteristics of rsFC in Patients with Acute MDD

#### 3.2.1. Comparison of rsFC between Baseline MDD and HCs

NBS identified a significantly decreased sub-network in the baseline MDD group compared to HCs (*p* < 0.001), including 137 ROIs and 197 edges (Appendix A, Figure 1A). We named this sub-network the “MDD reduced sub-network”. The highest-degree brain region (i.e., brain regions with the most connections) within this sub-network was the left precuneus (degree = 15), followed by left cerebellum cortex (degree = 14) and right superior temporal cortex (STG) (degree = 10).

NBS also revealed a significantly increased sub-network in the baseline MDD group (*p* < 0.001), composed of 99 ROIs and 131 edges (Appendix A, Figure 1B). We named this subnetwork the “MDD increased sub-network”. The highest-degree brain region was the left cerebellum cortex (degree = 17, x = 16.2, y = −47.2, z = −53), followed by the left precuneus (degree = 15) and the left cerebellum cortex (degree = 9, x = 36.7, y = −57.1, z = −32.8).

#### 3.2.2. Changes in Aberrant rsFC after 6-Month Treatment

We investigated the alterations in rsFC in the “MDD reduced sub-network” and “MDD increased sub-network” after the 6-month antidepressant treatment. Results revealed that, in 102 cases, rsFC in the “MDD reduced sub-network” remained significantly lower than those of HCs at follow-up, which showed state-independence (Appendix A). There were 42 cases where rsFC significantly increased after the 6-month antidepressant treatment, of which 40 exhibited no differences compared to the rsFC of HCs (Appendix A). Aside from this, there was no significant decrease in rsFC in the “MDD reduced sub-network” after the treatment (Appendix A).

In the “MDD increased sub-network”, in 61 cases, rsFC remained higher than those in the HCs at the 6-month follow-up, which showed state-independence (Appendix A). There were 31 cases where rsFC significantly decreased after the 6-month antidepressant treatment, of which 29 showed no differences compared to the rsFC of HCs (Appendix A). Besides, there was no significant increased rsFC in the “MDD increased sub-network” after the treatment (Appendix A).

RMANCOVA showed a significant effect of time on the overall strength of both sub-networks, with the “MDD reduced sub-network” increasing and the “MDD increased sub-network” decreasing after the 6-month treatment (Appendix A, Figure 2). *Post hoc* analysis showed that the overall strength of the “MDD reduced sub-network” was still lower than that of the HCs, and the overall strength of the “MDD increased sub-network” remained higher than that of the HCs.

#### 3.2.3. State-Dependence of the Aberrant rsFC

We investigated whether there was a significant interaction between time and remission status (remission vs. non-remission) on the rsFC in the “MDD reduced sub-network” and “MDD increased sub-network”. RMANCOVA showed a significant interaction between time and remission status on the rsFC between the left DLPFC and the left hippocampus (*p* = 0.0172, bold in Appendix A), as well as between the right OFC and the left angular gyrus (*p* = 0.009, bold in Appendix A). *Post hoc* analysis showed that the rsFC between the left DLPFC and the left hippocampus was significantly increased, and the rsFC between the right OFC and the left angular gyrus was significantly decreased in the remitters at 6-month follow-up, while no significant change was found in non-remitters. However, the interaction did not remain significant after FDR correction. Furthermore, there was no significant interaction between time and remission status on the overall strength of the two sub-networks.

### 3.3. Changes in rsFC before and after Treatment in Patients with MDD

We then used NBS to investigate the changes in rsFC before and after the antidepressant treatment in patients with MDD. NBS revealed a significantly increased sub-network after the 6-month treatment compared to baseline (*p* = 0.0012), which comprised 89 ROIs and 64 edges (Appendix A, Figure 3a). We named this sub-network the “follow-up increased sub-network”. The brain region with the highest degree was the right caudate lobe (degree = 11), followed by the right OFC (degree = 7).

NBS also found a decreased sub-network in patients with MDD after the 6-month follow-up compared to the baseline (*p* = 0.0136), composed of 61 ROIs and 36 edges (Appendix A, Figure 3b). We named this sub-network the “follow-up decreased sub-network”. The brain region with the highest degree was the right angular gyrus (degree = 5), and the right supramarginal gyrus (SMG) (degree = 5).

We analyzed the interaction of time by remission on the rsFC of the sub-networks above to figure out the relationship between the remission of depression and the changes in rsFC before and after the treatment. RMANCOVA revealed a significant interaction of time by remission on the rsFC in the “follow-up increased sub-network,” specifically between the right PFC and the left DLPFC (*p* = 0.020, bold in Appendix A), as well as between the right visual motor area and the left angular gyrus (*p* = 0.035, bold in Appendix A). However, none of these interactions were significant after FDR correction. RMANCOVA also revealed a significant interaction of time by remission on the rsFC in the “follow-up decreased sub-network,” specifically between the right Broca-pars opercularis and the right primary auditory area (*p* = 0.008, bold in Appendix A), between the right angular gyrus and the right primary auditory area (*p* = 0.001, bold in Appendix A), and between the right SMG and the left precuneus (*p* = 0.026, bold in Appendix A). The significant interaction of time by remission on the rsFC was only observed between the right angular gyrus and the right primary auditory area after FDR correction (*p* = 0.041).

### 3.4. The Overlap between Networks

To examine whether antidepressants could restore aberrant baseline rsFC in patients with MDD, we calculated the overlaps in sub-networks to examine whether antidepressants could increase the lower rsFC in MDD patients compared to HCs and whether antidepressants could decrease the higher rsFC in MDD patients compared to HCs. As hypothesized, there was a significant overlap between the “MDD reduced sub-network” and the “follow-up increased sub-network” (*p* = 5.62 × 10^−4^). Specifically, there were three instances of rsFC that overlapped in these two brain networks (Table 3). Similarly, there was a significant overlap between the “MDD increased network” and the “follow-up decreased sub-network” (*p* = 1.90 × 10^−4^). Specifically, there were two instances of rsFC that overlapped in these two brain networks (Table 3).

### 3.5. Comparisons of rsFC at Baseline and Pre- and Post-Treatment Changes in rsFC between Remitters and Non-Remitters

We used NBS to compare rsFC at baseline between remitters and non-remitters. ANCOVA revealed that there were not significantly higher (*p* = 0.371) or lower sub-networks (*p* = 0.663) in remitters compared to non-remitters.

We performed RMANCOVA to compare changes in rsFC between remitters and non-remitters. There was no significant interaction of time by remission on the sub-networks (sub-network with increased rsFC in the remitters compared to the non-remitters: *p* = 0.843; sub-network with decreased rsFC in the remitters compared to the non-remitters: *p* = 0.663).

## 4. Discussion

In the present study, we investigated aberrant rsFC and its state-independence and -dependence in patients with MDD. In addition, we examined the rsFC changes during follow-up and their overlaps with baseline aberrant rsFC. Differences in rsFC at baseline and rsFC changes during follow-up in remitters compared to non-remitters were also examined. We found that there was large-scale aberrant rsFC in patients with MDD, but most of them did not directly correlate with treatment effects, with the majority of the rsFC demonstrating state-independence. Antidepressants were observed to potentially modulate the rsFC originating from the caudate nucleus, the OFC, the right angular gyrus, and the right SMG. Nevertheless, this rsFC may not be associated with remission in MDD. Although little rsFC was found to be shared between aberrant MDD networks and follow-up altered networks, the overlap between the two networks showed significance.

### 4.1. Aberrant rsFC in Baseline MDD

Our study found significant differences in rsFC between patients with MDD and HCs at baseline, with the differences mainly located in the precuneus, cerebellum, and STG. Previous research has found that the precuneus is related to low self-esteem, a sense of inferiority [40,41], and the dysfunction of autobiographical memory [42] in MDD. Our previous study discovered that patients with MDD had reduced spontaneous activity in the precuneus [43]. The current findings are consistent with the results of our previous study, namely that patients with MDD exhibited a significant change in the strength of a large amount of rsFC originating from the precuneus. In addition, recent studies have shown that approximately half of the cerebellum cortex is involved in high-level cognitive and emotional function [44,45], and there is mounting evidence of significant differences in the rsFC originating from the cerebellum in patients with MDD [26]. Specifically, previous studies have reported decreased rsFC from crus II and Ⅰ, as well as lobule VIIA to frontoparietal network (FPN), in patients with MDD [45,46,47]. Therefore, the cerebellum may play an important role in the pathophysiology of MDD and aberrant rsFC originating from the cerebellum may contribute to the cognitive and emotional dysfunction observed in MDD. Lastly, it is worth noting that the STG plays a special role in the recognition of facial emotions within the distributed face-processing systems [48]. Therefore, the decreased rsFC originating from STG may indicate the disrupted emotion processing in MDD. Further research is needed to better understand the specific role of this aberrant rsFC in MDD.

### 4.2. States-Independence and -Dependence of the Aberrant rsFC

Our study found that after 6 months of antidepressant treatment, 102 instances of rsFC in the “MDD reduced sub-network” and 61 instances of rsFC in the “MDD increased sub-network” remained significantly different from those of HCs at follow-up. Approximately half of the rsFC still exhibited significant differences from HC, indicating that the aberrant rsFC observed in the baseline period of MDD persisted during follow-up and was state-independent. The state-independent aberrant rsFC patterns may reflect the underlying neural mechanisms of depression and contribute to the vulnerability and maintenance of the disorder. In addition, we found that only a small amount of the aberrant rsFC observed at baseline recovered to HC levels during follow-up, and only two instances of rsFC exhibited state-dependence with a significant interactive effect of time by remission. Specifically, the rsFC between the left DLPFC and the left hippocampus was significantly increased in the remitters but not in the non-remitters. This may reflect the restoration of top-down cognitive control in patients with MDD [49]. Additionally, the rsFC between the right OFC and the left angular gyrus was found to be involved in social cognition and top-down emotional regulation [50]. Thus, the increased negative rsFC may contribute to the recovery of the emotional and cognitive functions in patients with MDD. These two instances of rsFC may be important targets for antidepressant efficacy, but other independent studies with large sample sizes are warranted to confirm our findings.

### 4.3. Changes of rsFC before and after the Treatment in Patients with MDD

After 6 months of treatment, 64 instances of rsFC were significantly increased in patients with MDD, primarily originating from the caudate nucleus and the OFC. Additionally, 36 instances of rsFC were found significantly decreased in patients with MDD, mainly originating from the right angular gyrus and the right SMG. These changes in rsFC before and after treatment were not directly related to treatment efficacy, except for five instances of rsFC that were found to have an interactive effect of time by remission. However, the changes in this rsFC may reflect the specific changes in symptoms of depression, such as anhedonia or cognitive function. For example, a previous study found that reduced caudate volume in patients with MDD was associated with anhedonia symptoms [51]. Studies also found that increased rsFC in the caudate nucleus was related to the improvement of anhedonia in patients with MDD after antidepressant treatment [52,53]. Thus, our findings show that the increasing rsFC originating from the caudate nucleus could be associated with improvements in anhedonia in MDD. In addition, the OFC is essential for cognitive function, including response inhibition, flexible associative encoding, and emotion or value [54]. It could be that antidepressants improve cognitive function by increasing the rsFC from the right OFC. The angular gyrus plays a core region in the default mode network (DMN), which is associated with individual rumination and self-referencing in patients with MDD [55]. Our results suggested that antidepressants may improve such internal symptoms in MDD by decreasing rsFC originating from the angular gyrus. Additionally, the SMG is involved in the FPN network and is crucial for empathy [56] and inhibitory control during the cognitive reappraisal of negatively valanced stimuli [57]. Therefore, one interpretation is that antidepressants may improve negative cognition by decreasing rsFC originating from the SMG. However, further follow-up studies are necessary to investigate the role of antidepressants in brain networks and their effects on depression symptoms. In addition, we found that only five instances of rsFC were related to the treatment response in MDD. Our finding was consistent with previous studies, which identified large-scale brain network changes after antidepressant treatment, with only a few instances of rsFC found to be related to the remission of depressive symptoms [28]. Specifically, the increased rsFC between the right PFC and the left DLPFC, following treatment, may reflect the restoration of the deficit in self-valuation in MDD [58]. Previous studies have linked increased activity in the Wernicke and temporal cortex was related to the dysfunction in language and memory processing circuits and auditory verbal hallucination in depressive status [59,60,61]. Thus, the decreased rsFC between the right angular gyrus and the right primary auditory area, as well as between the right Broca-pars opercularis and the right primary auditory area, may indicate the restoration of language and memory processing in MDD. The SMG and precuneus are key components of the posterior DMN. Our study found a decrease in rsFC between right SMG and the left precuneus after treatment, which is consistent with previous studies which found that antidepressants could normalize the high activity in the posterior DMN [62,63,64]. Furthermore, we observed a decreased rsFC between the right visual motor area and the left angular gyrus after treatment. Similar to our findings, previous studies on patients who had taken antidepressants have reported decreased rsFC within and between the auditory network and the visual network [65].

Finally, there were no significant differences between the remitters and non-remitters in terms of changes in pre- and post-treatment rsFC. This could be due to the small sample size of patients at the end of the 6-month follow-up, especially the small number of unremitted patients, which may reduce the statistical power. Alternatively, the stringent threshold used in the analysis may cause false negative results. Finally, it is possible that similar rsFC changes occurred in both remitters and non-remitters groups at follow-up and NBS analysis could not detect the differences. Further studies with larger sample sizes and lower statistical thresholds are needed to confirm these findings.

### 4.4. The Overlap between the Networks

We observed a significant overlap between the aberrant sub-networks in patients with MDD and the sub-networks that were altered after antidepressant treatment. However, we found only a few instances of rsFC that were shared between these networks. Our study demonstrated that antidepressants can alter certain abnormal rsFCs. However, we found no direct correlation between these changes and treatment efficacy. This suggests that antidepressants may exert a non-specific effect, which could account for their limited efficacy in a significant portion of patients, as well as the delayed onset of therapeutic effects, which often take several weeks to appear. This “imprecise regulation of brain networks” by antidepressants highlights the need for further research to identify more precise targets for MDD treatment [66].

## 5. Limitations

However, there were some limitations to our study. First, we did not control the use of types of antidepressants, and patients were allowed to use hypnotic sedative drugs or other medications that may have affected the results. However, 88.7% of patients with MDD in our study received SSRIs. Moreover, previous studies have found that different SSRIs share stable neuroimaging biomarkers [67,68]. Therefore, differences in antidepressant types may not bring significant biases to our results. Second, there were a significant number of withdrawal patients during the follow-up, which could have introduced biases to the results. Third, we could not rule out the placebo effect in our current study. The rsFC changes pre- and post-treatment could be related to symptom change or other factors rather than the antidepressant effect. Moreover, some studies have reported limited short-term test–retest reliability of rsFC measurements [69,70], which could affect the detection of pre- and post-treatment changes. Therefore, more longitudinal studies are necessary to confirm our findings. Finally, we did not include follow-up MRIs of HCs in our study, and the differences in rsFC may be influenced by scanner noise. Further studies with control groups are needed to validate our findings.

## 6. Conclusions

Using NBS, we found large-scale aberrant rsFC in patients with MDD compared to HCs, with most of these differences showing as state-independent, while only a few showed state-dependent changes. We also observed a significant overlap between the baseline aberrant network and follow-up altered network, although only a little rsFC was shared between the two networks. Last, we did not find any significant differences in rsFC at baseline or in rsFC changes from baseline to post-treatment between remitters and non-remitters. Our findings indicate that most aberrant rsFC in patients with MDD exhibits state-independence, which may reflect underlying neural mechanisms contributing to the vulnerability and maintenance of the disorder. Although antidepressants may modulate these aberrations, they do not achieve their therapeutic effects by specifically targeting the aberrant rsFC. Future studies incorporating placebo groups and follow-up control groups are needed to develop more precise neuroimaging biomarkers for MDD and its treatment, as well as a better understanding of the underlying neural mechanisms of antidepressant treatment in patients with MDD.

## Figures and Tables

**Figure 1 brainsci-13-00705-f001:**
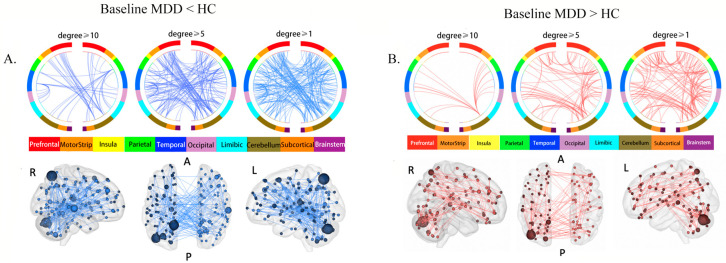
Aberrant rsFC between brain regions: (**A**) brain circle map and functional connectivity map of the “MDD reduced sub-network”; (**B**) brain circle map and functional connectivity map of the “MDD increased sub-network”. In the brain circle map (top row), different color represents different brain regions, and the three subgraphs show the rsFC in brain regions with degrees no less than 10, 5, and 1, respectively. In the brain functional connectivity map (bottom row), the size of the sphere represents the degree of the brain region (the number of rsFC connected to one brain region), and the larger the degree, the larger the sphere. Abbreviations: R: right view. A: anterior view. P: posterior view. L: Left view.

**Figure 2 brainsci-13-00705-f002:**
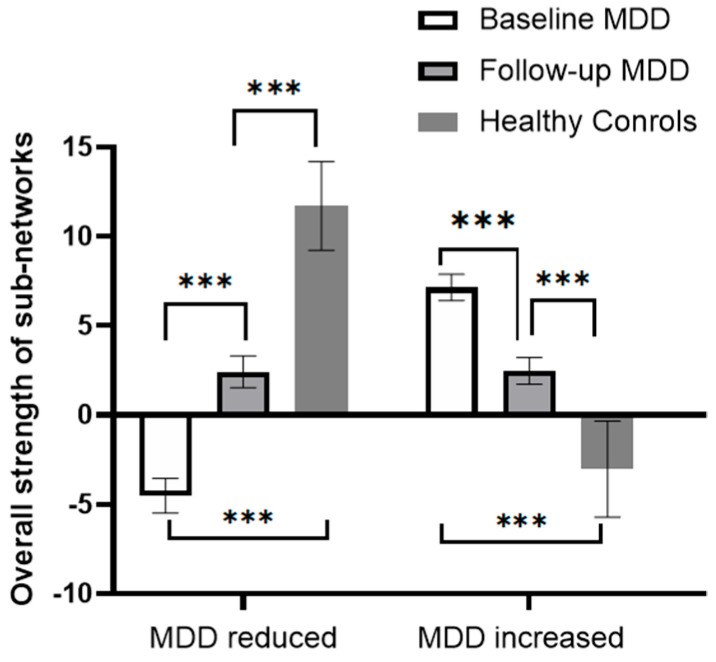
Changes in the overall strength of the sub-network in patients with MDD before and after 6 months of treatment. The bars in the figure represent the sum of the total strength of rsFC in the subnetworks, and error bars represent standard errors. *** *p* < 0.001.

**Figure 3 brainsci-13-00705-f003:**
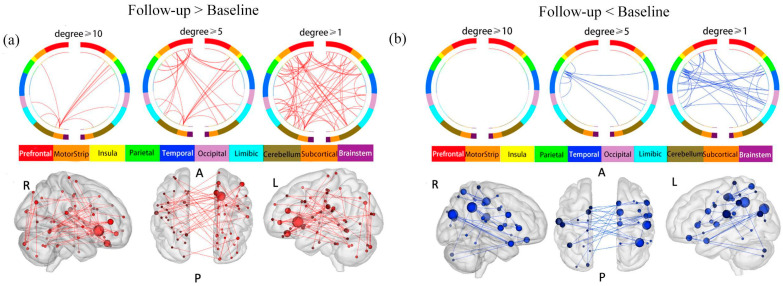
Changes in rsFC before and after treatment in patients with MDD. (**a**) Brain circle map and functional connectivity map of the “follow-up increased sub-network”. (**b**) Brain circle map and functional connectivity map of the “follow-up decreased sub-network”. Abbreviations and conventions as in Figure 1.

**Table 1 brainsci-13-00705-t001:** The difference of demographic and clinical characteristics among the baseline MDD, the follow-up MDD, the follow-up excluded, and HCs.

	Baseline MDD (*n* = 196)	Follow-up MDD (*n* = 114)	Follow-up Excluded (*n* = 82)	HCs (*n* = 143)	Baseline MDD vs. HCs		Follow-up MDD vs. HCs		Follow-up MDD vs. Follow-Up Excluded	
					*t/χ^2^* Value	*p* Value	*t/χ^2^* Value	*p* Value	*t/χ^2^/Z* Value	*p* Value
**Age (years, x ± s)**	35.9 ± 10.1	36.4 ± 10.3	35.3 ± 10.0	35.2 ± 9.6	0.697 ^a^	0.486	0.968 ^a^	0.334	0.722 ^a^	0.471
**Gender (%)**					1.849 ^b^	0.174	1.154 ^b^	0.283	0.056 ^b^	0.831
Male	120 (61.2%)	69 (60.5%)	51 (62%)	77 (53.9%)						
Female	76 (38.8%)	45 (39.5%)	31 (37.8%)	66 (46.1%)						
**Education (years, x ± s)**	10.3 ± 3.3	10.6 ± 3.4	9.7 ± 3.2	11.0 ± 3.5	1.826 ^a^	0.069	0.949 ^a^	0.344	1.880 ^a^	0.062
**Age of onset** **(years, x ± s)**	33.1 ± 7.0	32.7 ± 10.5	32.7 ± 10.5	-	-	-	-	-	0.750 ^a^	0.454
**Total illness duration, *M* (*P*_25,_ *P*_75_)**	18 (3, 65)	21 (6, 63)	16 (3, 75)	-	-	-	-	-	0.727 ^c^	0.476
**Current illness duration, *M* (*P*_25,_ *P*_75_)**	2 (2, 5)	2 (1.75, 4)	3 (2, 6)	-	-	-	-	-	1.796 ^c^	0.072
**Total number of episodes (x ± s)**	2.1 ± 1.4	2 ± 1.6	1.9 ± 2.1	-	-	-	-	-	1.196 ^a^	0.233
**Status of onset, (%)**					-	-	-	-	2.655 ^b^	0.103
**First onset**	80 (40.8%)	41 (36.0%)	39 (47.5%)	-						
**Recurrent onset**	116 (59.2%)	73 (64.0%)	43 (52.4%)	-						
**HAMD at baseline** **(x ± s)**	31.5 ± 6.9	31.5 ± 6.9	30.2 ± 7.2	1.4 ± 1.8	55.83 ^a^	<0.001	45.33 ^a^	<0.001	1.217 ^a^	0.225
**HAMA at baseline** **(x ± s)**	17.9 ± 6.3	17.8 ± 6.4	18.0 ± 6.0	1.3 ± 1.9	32.74 ^a^	<0.001	25.28 ^a^	<0.001	0.250 ^a^	0.803
**Antidepressants (%)**					-	-	-	-	5.441 *	0.067
Paroxetine	119 (60.7%)	78 (68.4%)	41 (50.0%)							
Other SSRIs	53 (27.0%)	23 (20.2%)	30 (36.6%)							
SNRIs	10 (5.1%)	8 (7.0%)	2 (2.4%)							
Others	14 (7.1%)	5 (4.4%)	9 (11.0%)							
**Combined medication (%)**										
Mood stabilizer	5(2.6%)	4(3.5%)	1(1.2%)		-	-	-	-	1.005 ^a^	1.306
Antianxiety	50(25.6)	29(25.4%)	21(25.6%)		-	-	-	-	0.533 ^a^	0.465
Antidepressant	23(11.7%)	15(13.2%)	8(9.8%)		-	-	-	-	0.023 ^a^	0.881
Antipsychotic	15 (7.6%)	9 (7.9%)	6 (7.3%)		-	-	-	-	0.023 ^a^	0.881
Sedative-hypnotics	94 (48.0%)	58 (50.9%)	36 (43.9%)		-	-	-	-	0.930 ^a^	0.335
**Baseline mean FFD (mm, x ± s)**	0.065 ± 0.042	0.061 ± 0.038	0.071 ± 0.046	0.062 ± 0.037	0.647 ^a^	0.085	0.298 ^a^	0.770	1.633 ^a^	0.029
**Follow-up mean FFD (mm, x ± s)**	-	-	-	-	-	-	-	-	-	-

^a^, *t* values obtained by independent-samples *T* test. ^b^, *χ^2^* values obtained by Pearson chi-square test. ^c^, Z values obtained by Mann–Whitney test. *, paroxetine and other SSRIs were combined for the Pearson test. Abbreviation: HCs, healthy controls. MDD, major depressive disorder.

**Table 2 brainsci-13-00705-t002:** The difference of demographic and clinical characteristics of participants among the remitters, non-remitters, and HCs.

	Remitters (*n* = 83)	Non-Remitters (*n* = 31)	HCs (*n* = 143)	Comparisons among Three Groups		Remitters vs. HCs		Non-Remitters vs. HCs		Remitters vs. Non-Remitters	
				*F/χ^2^* Value	*p* Value	*t/χ^2^* Value	*p* Value	*t/χ^2^* Value	*p* Value	*t/χ^2^/Z* Value	*p* Value
**Age (years, x ± s)**	36.6 ± 10.5	35.7 ± 9.8	35.2 ± 9.6	0.435 ^a^	0.648	1.054 ^a^	0.293	0.291 ^a^	0.771	0.411 ^a^	0.682
**Gender (%)**				1.661 ^b^	0.436	2.153 ^b^	0.142	0.051 ^b^	0.821	1.416 ^b^	0.234
Female	53 (63.9%)	16 (51.6%)	77 (53.9%)								
Male	30 (36.1%)	15 (48.4%)	66 (46.1%)								
**Education** **(years, x ± s)**	10.8 ± 3.4	9.9 ± 3.5	11.0 ± 3.5	2.527 ^a^	0.081	0.357 ^a^	0.721	1.527 ^a^	0.129	1.247 ^a^	0.215
**Age of onset** **(years, x ± s)**	33.0 ± 10.8	32.1 ± 9.7	-	-	-	-	-	-	-	0.392 ^c^	0.696
**Total illness duration *M* (*P*_25,_ *P*_75_)**	14 (3,70)	25 (13,61)	-	-	-	-	-	-	-	1.228 ^d^	0.219
**Current illness duration *M* (*P*_25,_ *P*_75_)**	2 (1, 4)	2 (2, 4)	-	-	-	-	-	-	-	0.340 ^d^	0.734
**Total number of episodes (x ± s)**	2 ± 1.7	2 ± 1.3	-	-	-	-	-	-	-	0.074 ^c^	0.941
**Status of onset (%)**				-	-	-	-	-	-	1.908 ^b^	0.167
**First onset**	33 (39.7%)	8 (25.8%)	-								
**Recurrent onset**	50 (60.3%)	23 (74.2%)	-								
**HAMD at baseline** **(x ± s)**	31.7 ± 7.1	30.9 ± 6.5	1.4 ± 1.8	267.5 ^a^	<0.001	38.27 ^a^	<0.001	25.19 ^a^	<0.001	0.561 ^a^	0.576
**HAMD at the end of 6 months (x ± s)**	2.4 ± 3	14.5 ± 9.0	-	-	-	-	-	-	-	-	-
**HAMA at baseline** **(x ± s)**	17.9 ± 6.6	17.6 ± 6.1	1.3 ± 1.9	220.1 ^a^	<0.001	16.61 ^a^	<0.001	14.27 ^a^	<0.001	0.241 ^a^	0.81
**HAMA at the end of 6 months (x ± s)**	1.6 ± 1.8	10.2 ± 5.7	-	-	-	-	-	-	-	-	-
**Antidepressants (%)**					-	-	-	-	-	0.444 *^b^	0.801
Paroxetine	59 (71.1%)	19 (61.3%)	-								
Other SSRIs	15 (18.1%)	8 (25.8%)	-								
SNRIs	6 (7.2%)	2 (6.5%)	-								
Others	3 (3.6%)	2 (6.5%)	-								
**Drug combination (%)**											
Mood stabilizer	3(3.6%)	1(3.2%)	-	-	-	-	-	-	-	0.0004 ^b^	0.983
Antianxiety	22(26.5%)	7(22.6%)	-	-	-	-	-	-	-	0.004 ^b^	0.951
Antidepressant	9(10.8%)	6(19.4)	-	-	-	-	-	-	-	2.222 ^b^	0.136
Antipsychotic	6 (7.1%)	3 (9.7%)	-	-	-	-	-	-	-	0.186 ^b^	0.666
Sedative-hypnotics	43 (51.8%)	15 (48.4%)	-	-	-	-	-	-	-	0.106 ^b^	0.745
**Baseline mean FFD (mm, x ± s)**	0.060 ± 0.039	0.065 ± 0.036	0.062 ± 0.037	0.251 ^a^	0.779	0.534 ^a^	0.608	0.314 ^a^	0.750	0.633 ^a^	0.529
**Follow-up mean FFD (mm, x ± s)**	0.068 ± 0.044	0.063 ± 0.036	-	-	-	-	-	-	-	0.524 ^a^	0.135

^a^, F-and *t*-values obtained by ANOVA and *post hoc* test, ^b^, *χ^2^* values obtained by Pearson chi-square test. ^c^, *t* values obtained by independent-samples *T* test. ^d^, Z values obtained by Mann–Whitney test. *, paroxetine and other SSRIs were combined for the Pearson test.

**Table 3 brainsci-13-00705-t003:** The overlaps between aberrant rsFC and the changes of rsFC after the treatment.

Brain Region 1	MNI Coordinate (x, y, z)			Brain Network	Brain Region 2	MNI Coordinate (x, y, z)			Brain Network
**The overlap between the** **“MDD reduced sub-network” and “follow-up increased sub-network”**									
R supramarginal gyrus	13.9	56.9	−16.6	Mot	R orbitofrontal gyrus	49	−58.1	14.4	DMN
L cerebellum	13.9	56.9	−16.6	SAL	R dorsolateral prefrontal lobe	61.8	−22.9	−22.4	FP
R secondary visual cortex	13.9	56.9	−16.6	VI	L primary sensory area	7.8	34.7	17.1	Mot
**The overlap between the** **“MDD increased network” and “follow-up decreased sub-network”**									
R orbitofrontal gyrus	5.1	34.9	−17.4	DMN	L visual associative cortex	−41.3	−75.4	22.8	DMN
R precuneus	7.5	−57.3	61.8	SAL	R dorsal posterior cingulate cortex	7.8	−23.1	44.9	Mot

L, left. R, right. FP, frontal-parietal. DMN, default mode network. Mot, motor network. VI, visual network I. SAL, Salience network.

## Data Availability

For more information on the data supporting the analyses and results, please contact the corresponding author, Yumeng Ju.

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
