# Peer review of "Aberrant Resting-State Functional Connectivity in MDD and the Antidepressant Treatment Effect—A 6-Month Follow-Up Study"

_brainsci, 2023, doi:10.3390/brainsci13050705_

Round 1

Reviewer 1 Report

Comments and Suggestions for Authors

In this study, the authors investigated aberrant resting state functional connectivity (rsFC) and their state-independence and -dependence in patients with major depressive disorder (MDD). The paper is well organized and well written.

However, there are some concerns to be addressed:

1.       The authors should consider underlining the novelty of the study in the Introduction.

2.       3.2.2. Changes in aberrant rsFC after 6-month treatment: it is not clear if the differences between the rsFC of before and after the treatment are linked to different treatment type. It is possible that different treatments lead to different effects? The authors should discuss this point more clearly. Or they might consider putting the treatment as covariate for the analysis.

3.       Is the overlap between the networks significative? Please consider including a figure that shows this overlap to underline your findings.

4.       Pag 12 “ Finally, there were no significant differences between the Remitters and Non-remit- 433 ters in terms of changes in pre- and post-treatment rsFC. This could be due to the small 434 sample size of patients at the end of the 6-month follow-up, especially the small number 435 of unremitted patients, which may reduce the statistical power.” Could the authors calculate the effect size Cohen’s d for supporting this theory?

Minor comments:

1.       Table 1 is too big and not very clear. Please consider improving it.

2.       The resolution of Figure 1 is poor. Please consider improving it.

Author Response

Response to Reviewer 1 Comments

In this study, the authors investigated aberrant resting state functional connectivity (rsFC) and their state-independence and -dependence in patients with major depressive disorder (MDD). The paper is well organized and well written. However, there are some concerns to be addressed:

Point 1: The authors should consider underlining the novelty of the study in the Introduction.

Response 1: Thank you very much for the suggestion. In response to your comment, we have made several revisions to highlight the novelty of our study in the Introduction, including how our study extends previous work in the field and how we could provide a new perspective on the topic. Firstly, we have added how our study extends previous work in the field (page 2, line 67-73): Previous findings regarding the neurobiological mechanisms underlying the normalizing of aberrant rsFC by antidepressants in the treatment of MDD have been inconsistent and heterogeneous, and many studies have only investigated specific regions or networks instead of the whole-brain. Therefore, there remains a need for further investigation into the effects of antidepressants on aberrant rsFC at the whole-brain level, in order to better understand the effects of antidepressants on aberrant rsFC and their potential role in the treatment of MDD. Secondly, some of the research questions we explored may offer new insights and perspectives that has not been fully understood by previous research (page 2, line 49-53): However, it remains unclear whether the aberrant rsFC are state-independent or state-dependent, as state-dependent rsFC are associated with the state of the depression and may serve as important mediators of the MDD outcomes [19], while state-independent rsFC may reflect the susceptibility and underlie the biological basis of the on-set and recurrence of MDD [20]. And (page 2, line 79-83) However, it should be noted that many studies on antidepressant treatment have been short-term and have not thoroughly investigated the long-term outcomes. Hence, there is a need for further exploration of biomarkers that may indicate sustained non-remission in depression.

Ponit 2: 3.2.2. Changes in aberrant rsFC after 6-month treatment: it is not clear if the differences between the rsFC of before and after the treatment are linked to different treatment type. It is possible that different treatments lead to different effects? The authors should discuss this point more clearly. Or they might consider putting the treatment as covariate for the analysis.

Response 2: Thank you very much for raising this question. We have acknowledged that the treatment type may potentially influence the rsFC changes observed in our study. However, in our sample, the vast majority (87.7%) of our patients received SSRIs as their antidepressant treatment (as shown in Table 1). Therefore, the differences between the rsFC before and after treatment may not be largely affected by the medication type in our study. We have also added this limitation in our discussion (page 12-13, line 463-468): First, we did not control the use of types of antidepressants, and patients were allowed to use hypnotic sedative drugs or other medications that may have affected the results. However, there were 88.7% of patients with MDD in our study received SSRIs. Moreover, previous studies have found that different SSRIs share stable neuroimaging biomarkers [68,69]. Therefore, differences in antidepressant types may not bring significant biases to our results.

Point 3: Is the overlap between the networks significative? Please consider including a figure that shows this overlap to underline your findings.

Response 3: Thank you very much for the thoughtful suggestion. The overlap between the networks is significant, and we have listed the overlapping rsFC in Table 3. There were 3 rsFC that overlapped in between the “MDD reduced sub-network” and the “follow-up increased sub-network”. Similarly, there were 2 rsFC that overlapped between the “MDD increased network" and the “follow-up decreased sub-network”. Despite the limited number of overlapping rsFC, their significance is notable. We attempted to use different colors to highlight the overlapping connections, while it was difficult to effectively visualize the overlapping rsFC due to the large number of rsFC in networks. As an alternative approach, we have listed the overlapping connections in a clear and organized manner in Table 3.

Point 4: Pag 12 “Finally, there were no significant differences between the Remitters and Non-remitters in terms of changes in pre- and post-treatment rsFC. This could be due to the small sample size of patients at the end of the 6-month follow-up, especially the small number of unremitted patients, which may reduce the statistical power.” Could the authors calculate the effect size Cohen’s d for supporting this theory?

Response 4: Thank you for raising this question. We have carefully considered your suggestion and believe that, in this particular case, calculating Cohen's d may not be very meaningful as we did not find any significant differences between these two groups. Instead, we have used g-power to calculate the sample size required to achieve statistical power. Results suggested that the minimum sample size to yield a statistical power of at least .8 (p<0.001) with a small effect size (d = 0.3) is 383, with a medium effect size (d=0.5) is 140, and with a large effect size was 57. Our sample size in the non-remitted group was only 31, which further validated our statement that the non-significant results between the remitted and non-remitted group was due to the small sample size.

Minor comments:

Point 1: Table 1 is too big and not very clear. Please consider improving it.

Response 1: Thank you for pointing this out. We have uploaded a clearer table.

Point 2: The resolution of Figure 1 is poor. Please consider improving it.

Response 2: Thank you for pointing this out. We have uploaded a high-quality figure.

Reviewer 2 Report

Comments and Suggestions for Authors

This manuscript studied the 6-month longitudinal changes of rsFC after antidepressant treatment in a group of MDD patients, identifying two significantly changed subnetworks in MDD after treatment in 6 months. In general, this is an interesting longitudinal study. There are a few concerns that need to be further addressed by the authors.

Introduction:

1. The rsFC is still not an established biomarker for MDD and related treatment, so the term "rsFC biomarkers" should be avoided in the manuscripts.

Methods:

2. For the fMRI data preprocessing, why didn't the authors remove the first 5-10 time point volumes? Why choose the low pass filter of 0.12 Hz? (while most of the resting-state use bandpass filter 0.01-0.1 or 0.01-0.08 Hz)

3. 20 out of 183 HC subjects were excluded due to large head motion. It's nearly 12%. what is the threshold of head motion for the fMRI scan? Which head motion parameters( and units) were present in the Tables? 

4. Please add a clear description for what're the criteria for the remitters and non-remitters. It seems there are no group differences in the clinical tests.

5. The authors conduct two directions of the contrasts in NBS analyses were tested separately, so the significant level for each tail of NBS test should be p < 0.025. 

6. Line 155, the four hypotheses should be null hypotheses.

Results:

7. Figure 2, the error bar style for 3 groups is different, please correct it. It seems the standard errors of MDD groups are smaller than the HC group, please also add the overall mean and std in the Table S2.

8. Why did the author just performed the repeat ANOVA analysis within the two change subnetwork instead perform an independent NBS to detect the state-related FC subnetwork, there might be one or more significant subnetworks that showed significant interactions of time and remission state.

9. what are the differences between the results in 3.2.3 and 3.3 since both are the results of the interaction between time and remission states? The author should just combine these two parts together. Also please clarify whether each result in these two sections is FDR corrected or not.

10. Why didn't the author perform the correlation analysis between the changes in rsFC and changes in clinical performance in the changed subnetworks? It would be very interesting whether those rsFC changes are related to the clinical evaluations.

11. Since there are kinds of literature showing low test-retest reliablities of RSFC over several weeks to months, the author should at least add this to the limitations.

Author Response

Response to Reviewer 2 Comments

This manuscript studied the 6-month longitudinal changes of rsFC after antidepressant treatment in a group of MDD patients, identifying two significantly changed subnetworks in MDD after treatment in 6 months. In general, this is an interesting longitudinal study. There are a few concerns that need to be further addressed by the authors.

Introduction:

Point 1: The rsFC is still not an established biomarker for MDD and related treatment, so the term "rsFC biomarkers" should be avoided in the manuscripts.

Response 1: Thank you for pointing this out, and we realized that the expression in our manuscript was inaccurate. We have revised this statement in page 2, line 86-87: to investigate the aberrant rsFC as potential biomarkers in MDD.

Methods:

Point 2: For the fMRI data preprocessing, why didn't the authors remove the first 5-10 time point volumes? Why choose the low pass filter of 0.12 Hz? (while most of the resting-state use bandpass filter 0.01-0.1 or 0.01-0.08 Hz)

Response 2: Thank you very much for raising this question. We did not remove the first 5-10 time point volumes as the removal of the initial volumes can also reduce the overall data volume and decrease the statistical power of the analysis (as we had only 180 time series). Additionally, recent studies have shown that removing the initial volumes or scrubbing may not always be necessary or beneficial (Parkes, L., Fulcher, B., Yücel, M., & Fornito, A. (2018). An evaluation of the efficacy, reliability, and sensitivity of motion correction strategies for resting-state functional MRI. NeuroImage, 171, 415-436.). In addition, we have included motion parameter as a nuisance regressor during the preprocessing step to control the influence of head motion. Therefore, we did not remove the first 5-10 time point volumes.

We chose to use a pass filter of 0.12 Hz, as this filter setting has been used in previous valid studies:  

  1. Rosenberg, M., Finn, E., Scheinost, D. et al. A neuromarker of sustained attention from whole-brain functional connectivity. Nat Neurosci 19, 165–171 (2016). https://doi.org/10.1038/nn.4179
  2. Yip, S.W.; Scheinost, D.; Potenza, M.N.; Carroll, K.M. Connectome-Based Prediction of Cocaine Abstinence. Am J Psychiatry 2019, 176, 156-164, doi:10.1176/appi.ajp.2018.17101147.
  3. Lake EMR, Finn ES, Noble SM.et al. The Functional Brain Organization of an Individual Allows Prediction of Measures of Social Abilities Transdiagnostically in Autism and Attention-Deficit/Hyperactivity Disorder. Biol Psychiatry. 2019 Aug 15;86(4):315-326. doi: 10.1016/j.biopsych.2019.02.019

By using the same filter setting, we can maintain consistency with the existing literature and ensure that our results are comparable to those of other studies.

Point 3: 20 out of 183 HC subjects were excluded due to large head motion. It's nearly 12%. what is the threshold of head motion for the fMRI scan? Which head motion parameters (and units) were present in the Tables?

Response 3: Thank you for bringing up this question. Firstly, the correct number of HCs should be 163 instead of 183. In our previous study with the same sample, we stated that: of the initial 163 HCs, 20 were excluded due to excessive head motion and incomplete basic information. We apologize for any confusion this may have caused and have revised this statement: Of the initial 163 HCs, 6 were excluded due to excessive head motion and 14 were excluded due to incomplete basic information or incomplete MRI data. In total, 143 HCs were included in the analysis. In addition, we have added the threshold of head motion “(mean Frame to Frame displacement, FFD, greater than 0.2 mm)” in page 2, line 135. And the head motion parameters present in the tables is mean FFD, which are expressed in millimeters. We have revised this in Table 1 & 2 accordingly (e.g., Baseline mean FFD (mm, x ± s)).

Point 4: Please add a clear description for what're the criteria for the remitters and non-remitters. It seems there are no group differences in the clinical tests.

Response 4: Thank you for your suggestion. We have revised the criteria for remitters to provide a clearer description (page3, line 141-142): Clinical remission was defined as a HAMD-24 score ≤ 7 at 6 month follow-up.

Point 5: The authors conduct two directions of the contrasts in NBS analyses were tested separately, so the significant level for each tail of NBS test should be p < 0.025.

Response 5: Thank you for pointing this out. Two directions of the contrasts in NBS analyses were tested separately, and one-tailed p value was derived so that p < .025 was taken as significant. We have revised as P < 0.025 in line 183 of page 4.

Point 6: Line 155, the four hypotheses should be null hypotheses.

Response 6: We thank the reviewer for pointing out this inaccuracy statement. We have added the word “null” in line 163 of page 4 accordingly.

Results:

Point 7: Figure 2, the error bar style for 3 groups is different, please correct it. It seems the standard errors of MDD groups are smaller than the HC group, please also add the overall mean and std in the Table S2.

Response 7: Thank you for pointing this out. We have adjusted the Figure 2 with the same error bars. And we have added a table (Supplemental Table S5) containing overall mean and std of each group.

Point 8: Why did the author just performed the repeat ANOVA analysis within the two change subnetwork instead perform an independent NBS to detect the state-related FC subnetwork, there might be one or more significant subnetworks that showed significant interactions of time and remission state.

Response 8: Thank you for your question. Actually, we have performed an independent NBS to detect the state-related FC subnetwork (kindly refer to result 3.5. Comparisons of rsFC at baseline and pre- and post-treatment changes in rsFC between Remitters and Non-remitters). And we found that there was no significant interaction of time by remission on the sub-networks (sub-network with increased rsFC in the Remitters compared to the Non-remitters: P = 0.843; sub-network with decreased rsFC in the Remitters compared to the Non-remitters: P = 0.663). Besides, the reason of conducting a repeated ANOVA analysis within the two subnetworks was to investigate the potential state-dependent or state-independent nature of the aberrant rsFC. Therefore, both independent NBS and the repeat ANOVA analysis within the two change subnetwork were conducted.

Point 9: what are the differences between the results in 3.2.3 and 3.3 since both are the results of the interaction between time and remission states? The author should just combine these two parts together. Also please clarify whether each result in these two sections is FDR corrected or not.

Response 9: Thank you for your suggestions. 3.2.3 investigated the interaction between time and remission status on the aberrant rsFC, aiming to investigate the potential state-dependence and -independence of these aberrant rsFC. On the other hand, in section 3.3, we examined the rsFC changes before and after antidepressant treatment, with the goal of assessing the modulation of antidepressant and their relation to treatment efficacy. As the aims of these two sections are different, we have decided to leave the sections as they are to accurately reflect our research objectives. Each result in both 3.3 and 3.2.3 is FDR correct.

Point 10: Why didn’t the author perform the correlation analysis between the changes in rsFC and changes in clinical performance in the changed subnetworks? It would be very interesting whether those rsFC changes are related to the clinical evaluations.

Response 10: Thank you for this thoughtful comment. Our analysis focused on examining the interaction effect between time and remission status on the changes of the aberrant rsFC. Consequently, we decided not to conduct a correlation analysis between clinical improvement and the changes of the aberrant rsFC to avoid redundancy in our analysis. 

Point 11: Since there are kinds of literature showing low test-retest reliabilities of RSFC over several weeks to months, the author should at least add this to the limitations.

Response 11: Thank you for your suggestions. We have mentioned this limitation in our discussion in page 13 line 470-473: Moreover, some studies have reported limited short-term test-retest reliability of rsFC measurements [70,71] which could affect the detection of pre- and post-treatment changes. Therefore, more longitudinal studies are necessary to confirm our findings.

Reviewer 3 Report

Comments and Suggestions for Authors

The article is very interesting and the topic is very important. I wonder if should not be placed in a more Psychiatry related journal, but I leave this decision to the authors and editor.

The figures in the pdf I saw are not really visible, making them bigger and of better quality.

Can you please discuss more the clinical usability of the findings?

How do you think these findings can be used in a clinical setting and with what degree of accuracy can be used to predict treatment response?

Any value in terms of a person-based approach? in other words, can you discuss more if the results can be used to stratify the MDD group at baseline and use the results for personalized treatment choice?

Depression has a cultural variability (that is also discussed in the DSM-5), how generalizable are the current findings in different geographical settings?

Thanks for this interesting study

Author Response

Response to Reviewer 3 Comments

The article is very interesting and the topic is very important. I wonder if should not be placed in a more Psychiatry related journal, but I leave this decision to the authors and editor.

Point 1: The figures in the pdf I saw are not really visible, making them bigger and of better quality.

Response 1: Thank you for pointing this out. We have uploaded a high-quality figure.

Point 2: Can you please discuss more the clinical usability of the findings?

Response 2: Thank you for your questions. Previous studies pointed out that baseline aberrant rsFC may help with clinical diagnose of MDD, differentiate between bipolar and unipolar depression and predict the treatment response [1]. Especially, state-independent rsFC, which is associated with MDD susceptibility, could serve as a reliable biomarker for MDD diagnosis. State-dependent rsFC may also play a crucial role in mediating MDD outcomes and can help clinicians choose appropriate treatments. However, the poor short-term test-retest reliability of rsFC limits its clinical use [2].

  1. Han, K.M.; De Berardis, D.; Fornaro, M.; Kim, Y.K. Differentiating between bipolar and unipolar depression in functional and structural MRI studies. Prog Neuropsychopharmacol Biol Psychiatry 2019, 91, 20-27, doi:10.1016/j.pnpbp.2018.03.022.
  2. Guo, C.C.; Kurth, F.; Zhou, J.; Mayer, E.A.; Eickhoff, S.B.; Kramer, J.H.; Seeley, W.W. One-year test-retest reliability of intrinsic connectivity network fMRI in older adults. Neuroimage 2012, 61, 1471-1483, doi:10.1016/j.neuroimage.2012.03.027.

Point 3: How do you think these findings can be used in a clinical setting and with what degree of accuracy can be used to predict treatment response?

Response 3: Thank you for your question. Regarding the prediction of treatment response, our results did not show significant differences in baseline rsFC between the remitters and non-remitters, indicating that our study could not distinguish the two groups at baseline. Many studies have used machine learning to predict treatment response [1-3]. The accuracy of machine learning models in predicting treatment response can vary depending on many factors, such as the type and complexity of the algorithm, the size and quality of the dataset, and the characteristics of the patient population. Some studies have reported high accuracy rates (e.g., above 80%) in predicting treatment response using machine learning models, while others have reported lower rates [3-4]. It is important to note that the prediction of treatment response should be validated in a prospective study in order to achieve accurate predictions that can aid clinical decision-making. However, such prediction of treatment response should not be the sole basis for clinical decision-making and should be interpreted in conjunction with other clinical information.

  1. Squarcina, L.; Villa, F.M.; Nobile, M.; Grisan, E.; Brambilla, P. Deep learning for the prediction of treatment response in depression. Journal of Affective Disorders 2021, 281, 618-622, doi:https://doi.org/10.1016/j.jad.2020.11.104
  2. Patel, M.J.; Khalaf, A.; Aizenstein, H.J. Studying depression using imaging and machine learning methods. Neuroimage Clin 2016, 10, 115-123, doi:10.1016/j.nicl.2015.11.003.
  3. Lee, Y.; Ragguett, R.M.; Mansur, R.B.; Boutilier, J.J.; Rosenblat, J.D.; Trevizol, A.; Brietzke, E.; Lin, K.; Pan, Z.; Subramaniapillai, M.; et al. Applications of machine learning algorithms to predict therapeutic outcomes in depression: A meta-analysis and systematic review. J Affect Disord 2018, 241, 519-532, doi:10.1016/j.jad.2018.08.073.
  4. Sajjadian, M.; Lam, R.W.; Milev, R.; Rotzinger, S.; Frey, B.N.; Soares, C.N.; Parikh, S.V.; Foster, J.A.; Turecki, G.; Müller, D.J.; et al. Machine learning in the prediction of depression treatment outcomes: a systematic review and meta-analysis. Psychol Med 2021, 51, 2742-2751, doi:10.1017/s0033291721003871.

Point 4: Any value in terms of a person-based approach? in other words, can you discuss more if the results can be used to stratify the MDD group at baseline and use the results for personalized treatment choice?

Response 4: Thank you for raising this questions. The results of the study cannot be used to stratify the MDD group at baseline for personalized treatment choice because no significant differences in baseline rsFC were found between remitters and non-remitters. Additional research is needed to replicate and to identify other biomarkers and clinical factors that can be used for treatment prediction.

Point 5: Depression has a cultural variability (that is also discussed in the DSM-5), how generalizable are the current findings in different geographical settings?

Response 5: Thank you for your questions. The generalizability of the current findings to different geographical settings may be limited by cultural variability in depression. We recruited participants in only one province in China, which may lead to relatively lower generalizability of our findings and implications. The symptoms of depression and the ways in which they are expressed and experienced can vary across cultures, which may impact the exploration of neural correlates of depression. Additionally, cultural differences in treatment-seeking behavior, access to treatment, and treatment preferences may also impact the generalizability of the findings. Therefore, it is important to replicate these findings in diverse populations and consider cultural factors when interpreting the results. It may also be necessary to conduct additional research to identify cultural differences in the neural correlates of depression and treatment response, in order to develop personalized treatments that are effective across different cultural settings.

Round 2

Reviewer 1 Report

Comments and Suggestions for Authors

The authors replied to all my concerns. I have no more questions.